# From the Gut to the Brain: The Role of Enteric Glial Cells and Their Involvement in the Pathogenesis of Parkinson’s Disease

**DOI:** 10.3390/ijms25021294

**Published:** 2024-01-20

**Authors:** Alba Montalbán-Rodríguez, Raquel Abalo, Laura López-Gómez

**Affiliations:** 1Department of Basic Health Sciences, Faculty of Health Sciences, University Rey Juan Carlos (URJC), 28922 Alcorcon, Spain; alba.montalban@urjc.es (A.M.-R.); laura.lopez.gomez@urjc.es (L.L.-G.); 2High Performance Research Group in Physiopathology and Pharmacology of the Digestive System (NeuGut-URJC), University Rey Juan Carlos (URJC), 28922 Alcorcón, Spain; 3Associated R+D+i Unit to the Institute of Medicinal Chemistry (IQM), Scientific Research Superior Council (CSIC), 28006 Madrid, Spain; 4Working Group of Basic Sciences on Pain and Analgesia, Spanish Pain Society, 28046 Madrid, Spain; 5Working Group of Basic Sciences on Cannabinoids, Spanish Pain Society, 28046 Madrid, Spain

**Keywords:** Parkinson’s disease, enteric glial cells, enteric nervous system, α synuclein, myenteric plexus

## Abstract

The brain–gut axis has been identified as an important contributor to the physiopathology of Parkinson’s disease. In this pathology, inflammation is thought to be driven by the damage caused by aggregation of α-synuclein in the brain. Interestingly, the Braak’s theory proposes that α-synuclein misfolding may originate in the gut and spread in a “prion-like” manner through the vagus nerve into the central nervous system. In the enteric nervous system, enteric glial cells are the most abundant cellular component. Several studies have evaluated their role in Parkinson’s disease. Using samples obtained from patients, cell cultures, or animal models, the studies with specific antibodies to label enteric glial cells (GFAP, Sox-10, and S100β) seem to indicate that activation and reactive gliosis are associated to the neurodegeneration produced by Parkinson’s disease in the enteric nervous system. Of interest, Toll-like receptors, which are expressed on enteric glial cells, participate in the triggering of immune/inflammatory responses, in the maintenance of intestinal barrier integrity and in the configuration of gut microbiota; thus, these receptors might contribute to Parkinson’s disease. External factors like stress also seem to be relevant in its pathogenesis. Some authors have studied ways to reverse changes in EGCs with interventions such as administration of Tryptophan-2,3-dioxygenase inhibitors, nutraceuticals, or physical exercise. Some researchers point out that beyond being activated during the disease, enteric glial cells may contribute to the development of synucleinopathies. Thus, it is still necessary to further study these cells and their role in Parkinson’s disease.

## 1. Introduction: Braak’s Hypothesis and the Enteric Nervous System

### 1.1. Parkinson’s Disease

Parkinson’s disease (PD) is the most common movement disorder and the second most common neurodegenerative disease, after Alzheimer’s disease [1]. It affects approximately 6 million people worldwide and its prevalence is expected to increase in the next few years [1,2]. In PD, an irreversible loss of dopaminergic neurons occurs in the substantia nigra and their corresponding axonal terminals in the striatum [1]. Most cases of PD are likely the result of a combination of environmental and genetic factors [3].

Symptoms are divided into motor and non-motor. Among the motor symptoms, bradykinesia, gait disturbances, tremors, rigidity, and speech deficits are the hallmarks of the disease. However, it is currently believed that PD begins many years before motor symptoms become clinically relevant [1], when some non-motor symptoms (depression, hyposmia (reduced sense of smell), cognitive impairment, sleep disorders, or constipation) may already occur [1,2].

In PD, the loss of dopaminergic neurons is associated with the development of Lewy’s bodies, protein inclusions containing disaggregated oligomers of many cellular proteins, mainly aggregates of α-synuclein (α-syn) [3,4]. Several studies have given different reasons for abnormal aggregation of α-syn [5]. For example, researchers have found mutations of the A30P, E46K, and A53T proteins, as well as duplications of the *SNCA* gene, which encodes α-syn. Also, oxidative stress is an important factor for the aggregation of α-syn to form Lewy’s bodies. Defects in ubiquitination have also been described as a cause for abnormal degradation of the protein by neuronal proteasomes [6,7,8].

### 1.2. Parkinson’s Disease and the Gastrointestinal Tract

Although α-syn aggregation in the substantia nigra is a key finding in PD, it also occurs in other body locations. Indeed, Braak’s hypothesis suggests that α-syn is firstly produced in the periphery and propagates to the central nervous system (CNS) to finally cause the disease [2,4]. One of the peripheral systems that stands out in support of the Braak’s hypothesis is the gastrointestinal (GI) tract. In the body, there is a constant communication between the CNS and the GI tract through the so-called brain–gut axis [9]. Thus, researchers have found a bidirectional spread of α-syn from the duodenum to the brainstem and the stomach after injecting α-syn in rats [10]. Another example of the validity of Braak’s theory is the study by Kim et al. who used a mouse model into which pathological α-syn fibrils were injected into the duodenal and pyloric muscular layer, demonstrating subsequent fibril spread to different regions of the brain [11]. Furthermore, it was observed that vagal nerve sections prevented the intestine-to-brain propagation of fibrils and neurodegeneration [11]. Thus, the vagus nerve seems to serve as the conduit for α-syn propagation from the gut to the CNS [2,4].

Despite obvious interspecific differences, human studies also support the gut-to-brain propagation underlying Braak’s hypothesis. For example, complete truncal vagotomy could be associated with a subsequent reduction in the risk of developing PD, preventing the spread of pathological α-syn to the brain [4,11,12]. Other human studies have focused on other locations where α-syn is also present like the appendix whose removal reduced the risk of developing PD [13].

Importantly, GI symptoms (hypersalivation, dyspepsia, constipation, abdominal pain, or defecatory dysfunction) are common non-motor manifestations of PD with at least one of them present in 60–80% of PD patients [1,2]. In support of Braak’s hypothesis, several research groups propose that these non-motor symptoms may indicate the onset of α-syn pathology in the intestine with inflammation and oxidative stress, later spreading to the brain [2,4]. Recently, it has been reported that in mouse models, α-syn is primarily expressed in terminals and varicosities of the intestine, where it modulates enteric neurotransmission and the development of cholinergic neurons. This has been linked to inflammatory processes in PD, and it was suggested that the change in intestinal microbial composition is the main source of misfolded (defective) α-syn in the intestine [2]. Furthermore, intestinal dysfunction, such as small intestinal bacterial overgrowth, irritable bowel syndrome (IBS), *Helicobacter pylori* infection, diarrhea, and inflammatory bowel disease (IBD) have been related to PD occurrence [4].

### 1.3. The Enteric Nervous System

To better understand Braak’s hypothesis and the role of the GI tract in PD, it is necessary to briefly describe the enteric nervous system (ENS), which is the largest and most complex subdivision of the peripheral nervous system [14,15,16]. During development, the cells derived from the neural crest invade, proliferate, and migrate within the intestinal wall, colonizing it and forming the origin of the enteric neurons and glial cells [15,16,17]. The ENS constitutes a neural network that, through “brain-like” neuronal circuits, provides local control over GI functions [18] such as motility and secretion/absorption, even in the absence of extrinsic innervation [15,19]. It also controls digestion and the elements of the GI defense barrier [19]. Due to this control, size, complexity, structure, and capacity to integrate information, the ENS is often referred to as “the second brain” [15,17,20,21].

Histologically, the ENS is organized into two plexuses: the myenteric plexus (or Auerbach’s plexus), which controls intestinal motility, and the submucosal plexus (or Meissner’s plexus). The submucosal plexus is more prominent in the small and large intestine, than in the stomach and participates in regulating intestinal secretion, especially in the small intestine [14,15,16,17,22,23,24]. Both plexuses have differences within ganglia, in intestinal regions and even among species. These differences can also be influenced by factors such as age, gender, circadian phase, or diseases [19].

In mammals, different populations of enteric neuronal cells and enteric glial cells (EGCs) are found within the ENS plexuses [14,22,24]. It is well known that loss, degeneration and/or functional impairment of enteric neurons can lead to enteric neuropathies. These alterations can range from subtle changes in the biochemical profile or connectivity within the neuronal network to complete loss of ganglionic segments in the gut in conditions such as Hirschsprung’s disease [25]. In high contrast with the enteric neurons, EGCs have received less attention. However, these cells are the main component of the ENS (in terms of their numbers) and are distributed throughout the whole GI tract [22,26,27].

Thus, the objective of this narrative review is to better understand the role and the changes that occur in EGCs during PD, as their participation has been less studied so far.

## 2. Methods

To achieve our aim, the features of EGCs will be first briefly described (Section 3), followed by a specific analysis of the available literature on the relationship between EGCs and PD (Section 4). For the search of relevant information on this relationship, we have used the specialized health science database PubMed. As inclusion criteria, we have considered both preclinical studies, using laboratory animals or cell cultures, and studies in humans, with no date limit and including both positive and negative results. As exclusion criteria, we have eliminated the articles in which, even though the subject was addressed, the changes in EGCs during PD were not specifically evaluated. Applying these criteria, we have found a total of 21 articles on the subject (15 for preclinical studies and 6 for studies in humans) that have been analyzed to extract the most relevant information (see Section 4).

## 3. Enteric Glial Cells

Within the enteric ganglia, EGCs are in direct contact with neuronal membranes [17,24]. EGCs do not produce or present myelin sheaths, are smaller than enteric neurons [24] and outnumber them, as has been observed in humans, where the EGC population in the myenteric plexus exceeds that of neurons by six/seven times. Nevertheless, this ratio varies by species [15,27]. For example, in the ileal myenteric ganglia of guinea pigs, the EGC population is only twice that of neurons [24].

EGCs exhibit extensive heterogeneity. Furthermore, they are phenotypically plastic and adapt their functions in response to environmental signals to maintain homeostasis [18,21]. This phenotypic plasticity is reflected in pathological processes such as neuroinflammation, cancer, and/or infections, as they can acquire proinflammatory or protumorigenic phenotypes [18].

Due to their significant role in regulating GI functions and their specific importance in pathological conditions, we will first explore the characteristics of these cells, their subtypes, markers, functions, their association with common GI disorders, and, finally, their involvement in PD.

### 3.1. Types of Enteric Glia

Currently, subpopulations of EGCs are categorized based on their morphology and anatomical location within the intestinal wall (Table 1). The different types of enteric glia vary both between regions of the digestive tract and among species. Furthermore, the expression of glial genes is influenced by factors such as circadian rhythms and age [18]. Nevertheless, it is still unknown how the morphology relates to adult EGCs in these areas and how the mature cell structure of plexus layers influences the phenotype of enteric glia [21].

Enteric glia express various receptors and constantly monitor their extracellular environment. This is why these cells are almost always activated and rarely at rest. Glial activation triggers a cascade of physiological intracellular signal transduction, such as those mediated by Ca^2+^ or cyclic adenosine monophosphate (cAMP). The effects of these cascades are generally beneficial and work to modulate intestinal reflexes and/or maintain homeostasis [18].

Depending on its state of activation, glia can be classified as activated, reactive or dysfunctional. Activated or reactive glia have physiological and defensive functions within the ENS, and dysfunctional glia may be responsible for enteric pathologies (Table 2).

### 3.2. Enteric Glial Markers

EGCs express various factors/proteins that can be used for their specific detection, as summarized in Table 3.

Studies have shown that the nuclear factor Sox-10 is involved in gliogenesis in the intestine [27], and its absence leads to total intestinal aganglionosis [21]. However, although most mature and immature EGCs express Sox-10, there is also a small population of EGCs that express glial fibrillary acidic protein (GFAP) and are negative for Sox-10. Furthermore, the role of Sox-10 in the early stages of EGC development appears to differ from its later role in EGC development [21]. From week 6 to week 11 of gestation, there is abundant expression of Sox-10 and the calcium-binding protein B (S100β) and very little expression of GFAP [21]. Other studies indicate that during the 12th week of gestation in humans, both S100β and GFAP markers seem to be independently regulated [34,36,37]. From this point onward, the S100β signal strengthens and fully develops after 18 weeks [34].

There is activation and plasticity of EGCs when an environment of inflammation or infection is mimicked [38]. Reactive gliosis accompanied by an increased intensity of labelling of GFAP and S100β was observed in the small intestine of mice, while the number of neurons decreased [34,38]. In addition, the systemic administration of lipopolysaccharide (LPS) led to an increased intensity labelling of S100β in the proximal colon, duodenum, and cecum after 24 h [39]. GFAP intensity is also increased in all these regions after 24 h [34,39]. Incorrect regulation of GFAP has been linked to scenarios of inflammation, Crohn’s disease (CD), colon cancer, IBD or intestinal ischemia [26,34,40].

Although these are the most common markers for EGCs, studies have found other interesting ones. The typical marker of astrocytes and oligodendrocytes, proteolipid protein 1 (PLP1), a proteolipid protein form of myelin, was unexpectedly and specifically expressed by enteric glia in the adult mouse intestine. A study demonstrated that most PLP1-positive cells also coexpressed S100β, but coexpression with GFAP was less abundant [32]. Later, it was noted that PLP1-positive glial cells were also present in the stomach and cecum of postnatal and adult human. It is unclear why enteric glia present this kind of myelin gene expression since they are non-myelinating glia [34].

### 3.3. Functions of the Enteric Glia and Their Role in Disease

In recent years, many functions of the enteric glia have been discovered that were previously unknown, further highlighting the importance of these cells (Figure 1).

First, EGCs perform essential functions to support the survival and functions of neurons in the ENS [27]. Among their traditionally recognized functions, they have a crucial role in supporting enteric neurons and providing nutrition to them, thus maintaining the body’s homeostasis [18,21,26,27]. Specifically, EGCs may provide metabolic support to the neurons of the ENS and also give structural preservation for ENS neurons and their axons during intestinal motility [27]. They also regulate oxidative stress by contributing to glutathione synthesis, an antioxidant agent that plays an important role in detoxification processes [18].

Another function of EGCs is the regulation of synaptic transmission, mediating communication between the nervous system and the immune system [18,21]. Furthermore, they participate in neurotransmission in the ENS and regulate reflexes and intestinal processes underlying neuroinflammation in the gut [18]. A series of studies have shown that EGCs are fundamental for regulating intestinal motility [18,21,26] and they extend their influence on intestinal epithelial cells, where glial factors affect the maturation and differentiation of the intestinal epithelium [18]. Also, it has been observed that the immune system affects GI motility, so the communication between both systems modulates intestinal functions such as motility, ion transport, and mucosal permeability. When the ENS is subjected to certain conditions, EGCs can act as an extension of the immune system [41]. EGCs are capable of detecting microorganisms and signals from immune cells through cytokine receptors. Thanks to this, they mobilize a cellular cascade that leads to the recruitment and activation of immune cells capable of eliminating an infection and repairing tissue damage [16].

EGCs have also been shown to be essential for secretion and absorption in the intestinal epithelium [26]. Furthermore, EGCs maintain the integrity of the epithelial barrier in the intestines that regulates interactions between luminal contents with the underlying immune system, and the rest of the body. It is also important to highlight that the glial-derived neurotrophic factor (GDNF) released by EGCs has an anti-inflammatory effect in the intestines by inhibiting cell apoptosis and reducing the levels of proinflammatory cytokines. When there is mild tissue inflammation, GDNF is involved in the processes of epithelial reconstitution and maturation [28].

Given the variety of functions performed by EGCs, they are implicated in multiple GI pathologies. In various immune disorders, it has been observed that EGCs respond to proinflammatory cytokines thereby increasing local inflammation [42]. EGCs could also present disease-related antigens to modulate adaptive intestinal immunity and influence barrier integrity [4,43]. In inflammatory conditions such as IBD, ulcerative colitis, CD or inflammation due to viral infections, the role of EGCs has been deeply demonstrated [44,45,46,47,48,49]. Another important GI disorder with changes in EGCs is IBS, a pathology that involves alterations in the brain–gut axis due to various factors, from previous inflammation of the gut to stressful conditions that lead to peripheral and central hypersensitization [50,51]. There are also changes in the EGCs in other conditions that affect the digestive system, including acute inflammation and eating disorders [52,53,54]. As mentioned above, some of these intestinal dysfunctions, particularly those displaying overt or low-grade inflammation, have been related to PD development [4]. Furthermore, neurological and neurodegenerative conditions, like Alzheimer’s disease, have also been associated with changes in gut functions and EGCs [52]. Thus, EGCs may also play a prominent role in PD, as discussed next.

## 4. Parkinson’s Disease and Enteric Glia

As mentioned above, Braak’s hypothesis is supported by clinical observations indicating an association between PD and less frequent intestinal movements (constipation) that already occur years before diagnosis and aggravate with PD progression [52].

Indeed, in PD, GI symptoms appear before central symptoms and the components comprising the ENS are affected by this condition. Changes in intestinal microbiota can cause disruptions in the function of the epithelial barrier and intestinal permeability, affecting the epithelial cells of the digestive tract, the immune system, and the ENS, including both neurons and glial cells. Alterations in these two enteric cell populations may contribute to the initiation of the misfolding of α-syn protein, which can lead to neurological dysfunctions extending along the brain–gut axis and cause PD [9]. Thus, although in PD both enteric dopaminergic neurons and EGCs are affected, it is the latter that, when undergoing changes, trigger harmful consequences for the body.

Interestingly, it has been shown that EGCs respond to damage activation through Toll-like receptors (TLR) 2 and 4. For example, it was demonstrated how TLR-4 knockout mice could avoid PD symptoms by suppressing the immune response mediated by EGCs [55]. This is particularly important in the case of the gut dysbiosis that is associated to PD [56]. As a consequence of the alteration in the microbiota presented by patients, the intestinal epithelial barrier (IEB) is altered, facilitating the entry of pathogens [57,58]. The potentially harmful bacteria inside the gut alter the epithelial barrier even more through the production of endotoxins like LPS, a structural component of the outer membrane of Gram-negative bacteria with immunostimulatory characteristics, which may provoke systemic inflammation and even sepsis [59]. Thus, LPS is recognized by TLR-4 and stimulates the secretion of a variety of pro-inflammatory cytokines that further affect gut epithelial barrier integrity, enhancing bacterial translocation in turn. In this context, EGCs respond to this kind of harmful stimulations through TLR-2 and TLR-4, protecting the host against pathogens [60]. However, reactive EGCs also produce proinflammatory cytokines, growth factors and other immunomodulatory molecules, such as nitric oxide, contributing to amplify the proinflammatory environment and increasing the damage of the intestinal barrier. Moreover, intestinal microbiota may be involved in the modulation of intestinal α-syn aggregation and may further play a role in PD [58,61] since bacterial components like LPS could increase α-syn aggregation, further supporting the idea that the gut microbiome is involved in PD initiation [57].

Thus, already in the early stages of PD, EGCs may trigger neuroinflammation, causing synaptic dysfunction and abnormalities in intestinal motility, due to their key role in regulating motility, intestinal permeability, and immune responses. The increase in proinflammatory markers such as interleukin-6 (IL-6), interleukin-1 (IL-1), tumor necrosis factor-alpha (TNF-α), GFAP, and S100β in the intestinal mucosa of PD patients underlies the functional findings [62].

There are studies showing that the colon of PD patients has a higher presence of the glial markers GFAP, S100β, and Sox-10. Furthermore, other studies have shown that patients with PD experience not only inflammation but also oxidative stress in the intestine due to biochemical alterations that disrupt the neuro–glia relationship in their ENS [58,63,64].

Taking all this into account, the 15 preclinical and 6 clinical studies that have specifically studied the possible implication of EGCs in PD are described in the following sections.

### 4.1. Studies Using Animal Models and Cell Cultures

Several preclinical studies have been carried out to clarify the relationship between PD and EGCs. The studies are very varied and have used different types of animal models (mouse, rat, primates) and cell cultures. Also, the methodologies used to provoke the symptoms of PD are very diverse. Information about these studies is summarized in Table 4.

#### 4.1.1. Rotenone and Other Pesticides

Mitochondrial dysfunction and oxidative stress are involved in the pathological mechanisms of PD. One of the most common animal models for PD studies is obtained by exposure to pesticides such as rotenone, a natural pesticide which is a mitochondrial complex I inhibitor. Long-term systemic administration of rotenone produces many features of PD in animal models [65,66,67].

One of the lines of research in this disease is interested in metalloproteins. Metallothioneins (MTs) released from astrocytes can protect dopaminergic neurons against oxidative stress. MTs are low-molecular-weight inducible proteins that bind to metals, such as zinc, copper, and cadmium, and contribute to detoxification by scavenging free radicals. The two major isoforms, MT-1 and -2, are expressed in most organs and show coordinated induction in response to various stimulants such as metals, hormones, cytokines, inflammation, and oxidative stress [67]. For that reason, some studies have examined the changes in MT levels by chronic systemic rotenone administration in the colonic myenteric plexus of C57BL mice. In addition, knockout (KO) animals have been used to elucidate the effects of MT depletion on rotenone-induced neurodegeneration in the ENS using MT-1 and MT-2 KO (KO) mice. In colonic sections obtained from wild-type C57BL mice, subcutaneous administration of rotenone for 6 weeks caused neurodegeneration and increased MT with EGCs activation (GFAP-positive glia) in the myenteric plexus calculated as integrated density (the sum of the values of the pixels in the image or selection). On the other hand, MT KO mice showed more severe myenteric neuronal damage after 4 weeks compared with the wild-type (WT) mice treated with rotenone, accompanied by reduced EGCs activation. In contrast to the upregulation of GFAP signals in WT rotenone-treated mice, rotenone markedly reduced myenteric GFAP immunoreactivity in MT KO mice. Moreover, 4-hydroxynonenal (4HNE) immunoreactivity, which is considered indicative of lipid peroxidation, was notably increased by rotenone administration in MT KO mice. These results imply that lipid peroxidation was aggravated by the deficiency of MT. In these animals, the myenteric neuronal damage was exacerbated by MT deficiency, accompanied by a reduction in the activation and protection performed by EGCs and an enhancement of oxidative stress. The findings also suggest that the presence of MT in EGCs protects myenteric neurons from rotenone-induced neuronal damage [65].

In 2015, the same research group analyzed changes in the activation of EGCs related with the duration of rotenone treatment (1-, 3- or 6-week administration of rotenone, 50 mg/kg/day). Rotenone significantly increased the integrated density of GFAP-immunoreactivity in the myenteric plexus at week six in the ascending colon of C57BL mice, but not in weeks one or three. The lack of activation of EGCs in the ENS at early stages may be involved in the rotenone-induced myenteric neurodegeneration. The results suggest regional differences in glial cell response to rotenone neurotoxicity, that may explain the temporal differences in the appearance and severity of neurodegeneration [66].

Searching for possible treatments that can alleviate the effects of this disease, the neuroprotective effects of the coffee compounds caffeic acid (CA) and chlorogenic acid (CGA) against rotenone-induced degeneration in C57BL/6J mice have been examined [67]. Animals were chronically administered with rotenone (2.5 mg/kg/day) for four weeks and orally administered CA or CGA daily for one week before rotenone exposure and during the four weeks of rotenone treatment. Administration of CA or CGA prevented rotenone-induced neurodegeneration of intestinal enteric neurons. CA and CGA upregulated the antioxidative molecules, MT-1,2, in striatal astrocytes of rotenone-injected mice, but not in EGCs. Authors did not detect any obvious MT signals in the intestine. Thus, it is still unclear whether MT is involved in the neuroprotective effects of CA and CGA against rotenone-induced enteric neurotoxicity in vivo. The neuroprotective effects and MT upregulation induced by CA and CGA were also analyzed in cultured enteric glial cells pretreated with CA or CGA for 24 h and with a low dose of rotenone (1–5 nM) for 48 h. MT-1,2 was expressed specifically in EGCs. Furthermore, CA or CGA prevented the rotenone-induced downregulation of MT in cultured cells. These findings suggest that CA and CGA could protect enteric neurons against rotenone toxicity by targeting the antioxidative properties of EGCs. Thus, in the enteric neuronal and glial co-cultures, rotenone treatment reduced MT-1,2 presence in glial cells and produced enteric neuronal loss, which were prevented by CA or CGA treatment [67].

The role of EGCs in PD with respect to mitochondrial function as well as the exact mechanisms of enteric glial activation are still largely unknown. Palanisamy and coworkers investigated the way in which EGCs respond to the pesticides present in the environment, like rotenone and tebufenpyrad, in both cell cultures and animal models [68]. When the rat immortalized EGC line CRL-2690 was exposed to rotenone or tebufenpyrad, cells decreased their metabolic activity, and mitochondrial function was impaired. Both pesticides provoked a reduction in mitochondrial mass, an increased mitochondrial fragmentation, and a reduction in the mitochondrial fusion protein mitofusin 2 in EGCs. These changes were associated with an imbalance in mitochondrial fission–fusion dynamics. In addition, the pesticides increased p62, an autophagy cargo adapter, and the autophagy protein microtubule-associated protein 1A/1B-light chain 3 (LC3). LC3 is an important protein in the autophagy pathway that participates in autophagy substrate selection and autophagosome biogenesis. Increasing LC3 levels imply an increased autophagosome formation. But p62 accumulates when autophagy is inhibited, and decreased levels can be observed when autophagy is induced [79]. For that reason, the lack of decreased p62 suggests an impaired autophagy-mediated protein degradation and thus an alteration of the autophagic processes. Accumulation of p62 and fragmented mitochondria may ultimately be a reason for neuronal cell death. Accordingly, in pesticide-treated cells, there was an upregulation of the proinflammatory proteins inducible nitric oxide synthase (iNOS), IL-6 and the TNF Superfamily Member 12, related with apoptosis, at the gene level. Palanisamy’s group also studied an animal model of PD induced in male Wistar rats by rotenone injection (2.8 mg/kg/day once daily for 4 days). In colonic samples, they did find an increase in the inflammatory marker iNOS in the myenteric plexus without changes in GFAP intensity. Thus, exposure to neurotoxic pesticides causes the impairment in mitochondrial bioenergetics and initiates inflammatory pathways in EGCs, which further enhances mitochondrial failure and proinflammatory processes leading to GI dysfunction [68].

An interesting study was performed in a C57BL/6J murine model of PD by oral rotenone treatment (10 mg/kg/day body weight) [55]. To more deeply explore the role of TLR-4 implication in PD-induced neuroinflammation, TLR-4-KO mice were used in parallel with oral administration of the pesticide rotenone, revealing reduced intestinal inflammation and intestinal and motor dysfunction, as well as less neuroinflammation and neurodegeneration. The presence of GFAP, as typical marker of enteric glial activity, as well as the accumulation of α-syn were studied in the colonic myenteric plexus. Analyses showed increased optical density of GFAP in the myenteric plexuses of WT rotenone-treated mice while TLR-4-KO mice were unaffected. Oral rotenone-treated mice also had an increased number of CD3+ T cells and TLR-4+ cells in the colonic mucosa, and this pro-inflammatory state was associated with an increase in GFAP positivity and α-syn pathology in the colonic myenteric plexuses. Taken together, these findings suggest that intestinal inflammation leads to activation of EGCs related to TLR-4-mediated gut inflammation in PD [55].

External factors such as stress can negatively influence the development of PD and should be considered in the study models. Stress-induced intestinal barrier dysfunction exacerbates the PD phenotype in the same PD model in C57BL mice, produced by oral administration of low doses of rotenone. For 12 weeks, mice received unpredictable restraint stress, and in the last six weeks, they also received the toxin (10 mg/kg/day). Immunofluorescence staining and optical density analysis of the optical density of GFAP revealed significant effects of the combination of stress and rotenone in the myenteric plexus, which synergistically promoted enteric glial activation [69].

The search for therapeutic targets to mitigate the progression of the disease is constant. Interestingly, the inhibition of the expression of the L-tryptophan-catabolizing enzyme tryptophan 2,3-dioxygenase (TDO) has been shown to inhibit aging-related α-syn toxicity. TDO catalyzes the first, rate-limiting step of the catabolism of the amino acid tryptophan in the kynurenine pathway, an enzymatic cascade responsible for the synthesis of nicotine adenine dinucleotide (NAD) and NAD phosphate. The kynurenine pathway of tryptophan metabolism produces neuroprotective as well as neurotoxic metabolites. Consistent with this, mice treated with a TDO inhibitor showed a decrease in rotenone-induced GFAP total fluorescence, as a marker of enteric glial cells activation, and decreased α-syn accumulation in the colonic myenteric plexus. These data support that TDO inhibition could be a potential therapeutic strategy to decrease motor, cognitive, and GI symptoms in PD [70].

#### 4.1.2. MTPT1-Methyl-4-phenyl-1,2,3,6-tetrahydropyridine (MPTP)

The neurotoxin 1-methyl-4-phenyl-1,2,3,6-tetrahydropyridine (MPTP), which selectively destroys cells in the substantia nigra, is one of the most extensively used neurotoxins to produce parkinsonism in animals [80]. MPTP is metabolised to the 1-methyl-4-phenylpyridinium ion (MPP (+)) in glia, and this ion is introduced into neurons via the dopamine transporter, increasing the oxidative stress in neurons. The mechanism used by MPP(+) to promote cell death is believed to involve redox-active metals, in particular, iron (Fe) [81].

Using this model of PD, a study is interested in characterizing the neurochemical coding of the ENS in the colon [71]. For this, MPTP-treated monkeys (*Macaca mulatta*) were used. Repetitive administration of MPTP over time in monkeys triggers a neurodegenerative process similar to humans with PD. The study of whole mount preparations of the myenteric plexus from the ascending colon did not reveal any significant difference in the number or in the phenotype of Sox-10-positive EGCs between the MPTP-treated model and controls, suggesting that EGCs are not a primary target of MPTP in the ENS. However, the authors found that the treatment produced a significant increase in the number of neurons in the myenteric plexus, while the density of EGCs did not change. As there are fewer glial cells around the neurons, they may be more exposed and less protected against infections or oxidative stress [71].

In contrast, another study carried out to characterize functional and neurochemical changes in the isolated ileum of the MPTP-treated marmoset showed different results [72]. Contrary to the study mentioned above [71], immunohistochemical analyses of the myenteric plexus showed that the number of the enteric glia labelled with Sox-10 increased, suggesting that MPTP treatment led to inflammation of the ileum [72].

The chronic MPTP/probenecid (MPTP/p) model is an improvement of the acute and subacute models using only MPTP and has an advantage in its ability to allow for the exploration of PD progression and mechanisms. The uricosuric agent probenecid co-administered with the dopaminergic neurotoxin MPTP produces a chronic mouse model of PD and serves to elevate concentrations of MPTP in the brain by reducing renal elimination of the toxin [82]. Chronic MPTP/p administration induced primary PD symptoms and neuroinflammation and increased α-syn levels in male C5BL/6 mice [73]. Mice received 10 doses of MPTP (25 mg/kg) in combination with probenecid (250 mg/kg). An acute model was also prepared with a single injection of MPTP (40 mg/kg) and probenecid. Chronically, an important increase in aberrant aggregated and nitrated α-syn was detected in the neurons and GFAP-positive EGCs in the gastric myenteric plexus. Moreover, α-syn inclusion bodies are only found in the chronic MPTP mouse model, but not in the acute or subacute models. In addition, they studied possible mechanisms in mice acutely injected with MPTP. The stomach naturally has elevated monoamine oxidase B (MAO-B) activity and reduced superoxide dismutase (SOD) activity, thereby increasing its susceptibility to MPTP-induced oxidative stress. A significant increase in reactive oxygen species (ROS) in the stomach and an elevated level of 4-HNE were found in EGCs 3 h post MPTP exposure. These results suggest that EGCs could be the initial cells contributing to synucleinopathies in the stomach [73].

#### 4.1.3. 6-Hydroxydopamine (6-OHA)

The neurotoxin 6-hydroxydopamine (6-OHDA) is widely used to develop models of PD in animals. Intracerebral infusion of 6-OHDA produces a massive destruction of nigrostriatal dopaminergic neurons that allows for the investigation of motor and biochemical dysfunctions in PD [83].

A study examined the impact of central dopaminergic degeneration, induced by intranigral injection of 6-OHDA and the effects on the distal colon of male Sprague-Dawley rats. After this, animals presented bowel inflammation, corroborated by an increase in eosinophil and mast cell density within the colonic mucosa and submucosa layers, associated with increased oxidative stress, an increase in pro-inflammatory cytokine levels, and enhancement of colonic excitatory tachykinergic neurotransmission and subsequent increased motility. Furthermore, EGCs of colonic myenteric ganglia displayed increased GFAP immunopositivity (evaluated as a percentage of positive pixels), related to inflammation [74].

In 2020, the same group using the same PD model observed a pathological remodeling occurring in the colon of 6-OHDA-treated rats, with significant alterations as compared with controls, with mild inflammation (eosinophil infiltration) and a transmural deposition of collagen fibers. Epithelial cells displayed a reduced claudin-1 and transmembrane 16A/Anoctamin 1 presence; goblet cells increased their mucin production; colonic crypts were characterized by an increase in proliferating epithelial cells. In this context, the density of S100β-positive glial cells (evaluated as cell number per analyzed area) was increased as well, suggesting enteric glial activation in the colon from PD patients [75].

Using this model, the inflammatory response during short periods after PD induction has been investigated. EGCs present different responses in the different layers of the intestine depending on time. Earlier than 48 h, authors detected an increase in GFAP in the muscle wall, suggesting that it was a primary event for the upregulation of GDNF, TNF-α, and occludin in the intestinal mucosa, which was observed after 1 week. The authors suggested that EGCs may be recruited from the myenteric plexus to the mucosa layer. After 2 weeks from PD model induction, there was no alteration in the fluorescence intensity level of GFAP protein, but there was a decrease in occludin. These modifications may be related with an early enteric signalization modulated by EGCs after parkinsonian neurodegeneration, followed by inflammatory signals. In general, 4 weeks post-injection, EGCs appeared to be modulated in both muscular and mucosal layers. The dual response that EGCs can develop demonstrates that they are plastic cells, which can either promote inflammation or intestinal tissue protection [76].

#### 4.1.4. Adeno-Associated Virus (AAV)-α-Synuclein

Intranigral delivery of adeno-associated virus (AAV)-α-synuclein induces widespread overexpression of human α-syn in the nigrostriatal pathway, both at the mRNA level and the protein level [84].

The effects of bilateral nigral administration of AAV-α-synuclein on gut microbiome, enteric neurons, and enteric glia has been studied in adult male Sprague-Dawley rats, combined with the impact of voluntary exercise in these animals. Physical exercise is related with a mitigation of the symptoms of PD and with changes in the gut microbiota. For this, rats were housed in either standard housing cages or in cages with free access to running wheels. Whole-mount preparations of proximal duodenum, ileum, and colon tissue were prepared to isolate the submucosal and myenteric plexuses, and immunofluorescence was performed. In this model of PD, the rats presented a significant neuronal loss in the ileal submucosal plexus with no change in enteric glia. In contrast, the myenteric plexus exhibited a significant increase in glial cells, but changes in the number of neurons were not detected. Accompanying alterations in the intestinal microbiome and bile acid metabolism were noted. Voluntary running protected from neuronal loss and from the increased EGCs in the PD model with brain injection of AAV-α-syn. In sedentary animals, AAV-mediated overexpression of α-syn significantly increased S100β intensity in the myenteric plexus compared to controls, which suggests increased gut inflammation. AAV-α- synuclein and control groups with voluntary running had a similar intensity of S100β-immunostaining. Thus, the development of nigral α-syn pathology in this model of PD provokes important alterations on the ENS that are attenuated by exercise [77].

#### 4.1.5. A53 α-Synuclein Mouse Model

The A53T α-synuclein mouse model is a genetic model of α-synucleinopathy which presents an accumulation of insoluble and aggregated α-syn in enteric neurons in both myenteric and submucosal plexus along with colonic motor abnormalities, characterized by an impairment of cholinergic neurotransmission in the early stages of α-syn-driven pathology, without concomitant CNS involvement [85].

The occurrence of enteric inflammatory responses in this model in the early stages before the onset of brain pathology has been examined [78]. For this, the researchers used pre-symptomatic transgenic mice at 3, 6, and 9 months of age. They found an increase in colonic IL-1β and TNF levels as well as enteric glia activation since 3 months of age. An increase in GFAP intensity in the mucosa, submucosa, and myenteric plexus was observed in mice at 3 months of age, thus suggesting the activation of EGCs. An increased co-localization of GFAP-positive glial cells and TLR-2 labelling was also observed in these animals. Of interest, TLRs, including TLR-2, are expressed on immune/inflammatory, intestinal epithelial cells as well as on enteric neurons and glial cells, and are involved in the triggering of immune/inflammatory responses and in the maintenance of intestinal barrier integrity, as well as in the shaping of gut microbiota [78].

### 4.2. Studies Using Human Biopsies from PD Patients

Some authors have conducted studies using samples obtained from patients with PD, summarized in Table 5.

Phosphorylated alpha-synuclein (p-α-syn) is found in the CNS of patients with PD, causing the development of Lewy’s bodies and Lewy’s neurites, the main neuropathological hallmarks [91]. α-syn deposits have been found in the GI tract of parkinsonian patients. These observations have suggested that mucosal biopsies obtained by endoscopy can help in the molecular diagnosis of PD. However, the potential utility of mucosal biopsies is not restricted to the detection of α-syn, but also expands to other crucial factors underlying the pathophysiologic mechanisms of GI symptoms in PD [92]. Nevertheless, some studies showed that samples of healthy controls were frequently positive for p-α-syn, thereby questioning its role as a biomarker for PD [93]. Even though the mere presence of p-α-syn-positive aggregates in SNE should not be considered a criterion for the diagnosis of PD, an elaborate morphometric analysis of p-α-syn-positive aggregates in GI biopsies could serve as a suitable tool for the in vivo diagnosis of PD [91].

One of the human studies that first revealed changes in EGCs was conducted in 2013 [63]. It is worth noting that in this work, the number of samples was very limited, as only four biopsies from the ascending colon from controls and PD patients were analyzed. The mRNA levels of pro-inflammatory cytokines (TNF-α, interferon-γ, IL-6, and IL-1β) and glial markers (GFAP, Sox-10, and S100β) were analyzed in two of these biopsies using real-time PCR. The other two samples were used for immunohistochemical analysis of p-α-syn to detect Lewy’s pathology in whole-mount preparations from submucosa. Proinflammatory cytokines, GFAP, and Sox-10 were significantly elevated in colonic samples from PD patients analyzed by real-time PCR. These changes were negatively correlated with disease duration but not with the presence of Lewy’s pathology. However, no significant changes were found in the presence of S100β, which have been seen in other studies, although the number of biopsies used may have been a limitation in this aspect. These findings may provide evidence that intestinal inflammation occurs in PD and support the role of peripheral inflammation in the onset and/or progression of the condition and also shows that enteric inflammation in PD is closely related to glial dysregulation [63].

To determine whether the enteric glia in PD becomes reactive, the changes in one of the fibrous proteins that form the intracellular intermediate filaments (GFAP) have also been studied [86]. The researchers assessed the presence and the levels of phosphorylation of GFAP in colonic biopsies of patients by Western blot. The assembly of GFAP is controlled by its phosphorylation state, as its soluble phosphorylated form is in dynamic equilibrium with the polymerized, non phosphorylated fraction of the protein. Thus, phosphorylation of GFAP at its amino-terminus residues causes disassembly of the intermediate filaments, and conversely, its dephosphorylation restores its potential to assemble [94]. The integrity of the cytoskeleton is essential for normal cell function. For that reason, GFAP phosphorylation may be an important regulatory mechanism. Contrary to previous observations of brain damage, hypophosphorylation of GFAP at serine 13 was found in enteric glia during PD. These findings seem to indicate that glial cells respond differently depending on the neurodegenerative process. In this study, patients with PD presented higher GFAP levels in their colonic biopsies, compared with control subjects at both mRNA and protein levels, in mucosal and submucosal EGCs, which seems to indicate a reaction of these cells to the damage. Subsequent additional experiments were conducted using real-time PCR to analyze the different GFAP isoforms found in EGCs. The results revealed that GFAPκ was the main isoform in colonic EGCs relative to GFAPα and GFAPδ. Interestingly, this isoform has a low propensity to form homomeric intermediate filaments, but the specific role of the elevated presence of this isoform in the physiology and function of EGCs remains to be determined. In addition, it remains to be determined whether other types of enteric glia such as those populating the myenteric plexus, whose study is less accessible in routine GI biopsies, also preferentially present GFAPκ like their mucosal and submucosal counterparts. In summary, the authors found that an enteric glial reaction occurs in patients with PD. The reactive gliosis in PD was associated with a drop in GFAP phosphorylation [86].

Alterations in the colonic mucosal barrier as well as changes in inflammatory markers in plasma and fecal samples of 19 PD patients have also been evaluated [87]. In this study, S100β was used as a glial cell marker to evaluate enteric glial activation on colonic mucosal specimens collected during colonoscopy. The reduction in epithelial neutral mucins and claudin-1 and the increase in acid mucins and collagen fibers was histologically detected, and, in contrast to what was observed by Devos, an increase in the number of S100-positive glial cells was also demonstrated. In these studies, authors also found both enteric inflammation and activation of EGCs. This activation can affect the integrity of the intestinal epithelial barrier, and provoke an inflammatory response, with abnormal tissue repair with development of fibrosis and increased synthesis of collagen fibers in the mucosa of PD patients [87].

Alterations provoked in the GI tract by PD have negative consequences for patients. One of the most important GI symptoms is the development of constipation. GDNF produced by EGCs is prominent for the morphogenesis of the ENS and has a key role in preserving the integrity of the mucosa, under the control of enteric glia. The relationship between serum levels of GDNF and constipation was analyzed in 128 PD patients using ELISA assay. Patients were divided into three groups: with PD but without constipation, with prodromal stage constipation, and with clinical stage constipation. The results were analyzed with binary logistic regression and suggested that GDNF may act as a protective factor in the prevention of constipation, since serum levels were lower in constipated patients with PD. The reduction in GDNF affects the integrity of the intestinal mucosal barrier, making its repair more complicated and leading to changes in intestinal permeability [88]. Inside the GI tract, enteroendocrine cells are part of the intestinal epithelium and are oriented towards the lumen. These cells are connected to enteric nerves and have been identified as a possible pathway by which pathological α-syn propagates to the myenteric plexus [95]. In addition, as mentioned above, it is increasingly accepted that the gut microbiota may provide insights into the understanding of the pathogenesis and response to treatment of PD patients [1,96]. Changes in permeability could affect the equilibrium relationship between all these elements and may be related to the spread of the disease from the GI tract to the CNS and contribute to the development of constipation.

In another study, the presence of leucine-rich repeat kinase 2 (LRRK2) in the ENS was characterized [89]. The *LRRK2* gene is a gene commonly associated with PD. Mutations in the *LRRK2* gene, which encodes leucine-rich repeat kinase2 (LRRK2), are a cause of autosomal dominant PD. The expression of LRRK2 is regulated by phosphorylation. LRRK2 expression was measured in full thickness samples of colon obtained from 16 Lewy’s body, 12 non-Lewy’s body cases, and 3 controls without neurodegeneration. This study was the first to show that EGCs express LRRK2. In human samples, immunoreactive cells were observed over the entire thickness of the ganglia and expressed in the cytoplasmic perinuclear region of both neurons and EGC. To complete the study, they performed fetal and adult EGCs cultures obtained from rat. Both types of cell cultures expressed LRRK2 protein. Additionally, the authors identified the cAMP pathway as a key signaling pathway involved in the regulation of LRRK2 expression and phosphorylation in EGCs. Thus, EGC-expressed LRRK2 could participate in modulation of intestinal α-syn aggregation and inflammation in the gut. It opens new perspectives on the possible role of LRRK2 in GI physiology and should prompt further studies of enteric glial LRRK2, which could become a new target for modulation of the intestinal tract functions and intestinal disorders [89].

Recently, a study has investigated α-syn disturbances and glial responses in duodenal biopsies from patients with PD [90]. The study was conducted in 18 patients with advanced PD, 4 untreated patients with early PD, and 18 healthy controls. Immunohistochemistry was made for anti-aggregated α-syn (5G4) and GFAP antibodies. A semiquantitative and morphometric analysis was then performed on the duodenal mucosa and submucosa to characterize α-syn-5G4+ and GFAP-positive cells density and size. Prior to performing immunohistochemical labeling of EGCs, the authors validated GFAP as a marker for enteric gliosis by performing double-immunofluorescent staining with S100β and Sox-10. They found that GFAP represents a broad marker for EGCs in the human duodenal mucosa and submucosa [26]. The presence of α-syn was detected in the duodenal biopsies of all patients with early and advanced PD. The immunohistochemical labeling for α-syn was localized in neurons. In addition, evaluation of EGCs demonstrated an increased size and density suggesting reactive gliosis. However, in this case, authors did not find colocalization between α-syn-5G4 and GFAP markers [90].

## 5. Conclusions

In general, the studies described here show an activation of EGCs associated with PD and an increase in their number in response to inflammation, most likely with a protective role of neurons against damage. Furthermore, different external factors like stress can affect EGC activation and exacerbate PD symptoms. Based on all this, diverse potential therapeutic strategies have been proposed, such as the inhibition of TDO [70], voluntary running [77], or the use of polyphenols derived from coffee [67]. Other nutraceuticals, mainly those with antioxidant activity, might also be useful to modulate EGCs in PD, because they seem to be beneficial in other pathological conditions [28,35]. Another possible target to consider could be the modulation of the endocannabinoid system, since cannabinoids and cannabinoid-like compounds can regulate EGC activity, directly through peroxisome proliferator-activated receptors (PPAR)α receptors by palmitoylethanolamide and indirectly through activation of PPARγ or cannabinoid type 2 receptors (under GI inflammatory conditions) [97]. Moreover, the fact that dysbiosis is present in PD patients points to the possibility of treatment with probiotics, prebiotics, or combinations (synbiotics), and others such as postbiotics or paraprobiotics [98].

Interestingly, some authors go further and suggest that EGCs may have a relevant role in the development of the disease itself and contribute to the onset of α-synucleinopathies [73,89].

Nevertheless, there is still no solid causal evidence of a direct relationship between EGCs since some reports failed to show this role. For example, studies in macaques found no changes in the Sox-10 marker in a PD model [71]. In human studies, there is an important heterogeneity in GFAP levels between PD patients, and levels of glial markers have been negatively related to the duration of the disease [63].

In conclusion, the gathered information highlights the importance of EGCs in PD, suggesting they may have a significant role in the pathogenesis and its GI symptoms, and opening new doors for future investigations. However, it seems that we are still far from comprehensively understanding the mechanisms and ways of preventing the disease before patients begin to develop it. For that reason, it is still necessary to study these strategic cells more deeply, in the hope that these studies will help improve treatment and prevention of a neurodegenerative disease that affects an increasing number of people every day.

## Figures and Tables

**Figure 1 ijms-25-01294-f001:**
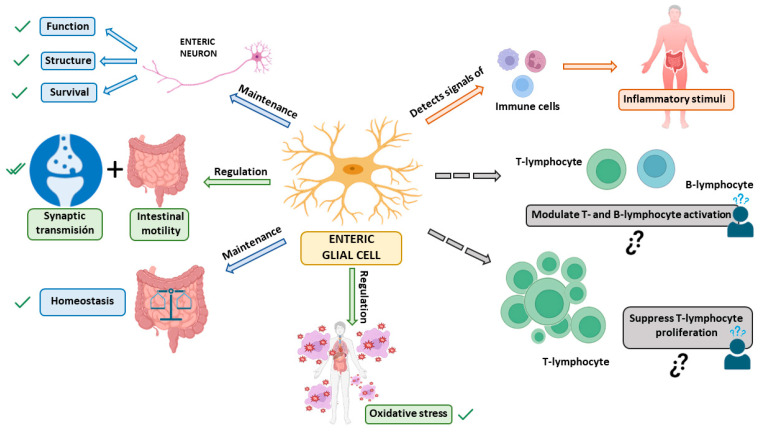
Functions of enteric glial cells (EGCs). The main functions of EGCs are: 1—maintenance of structure, function, and survival of enteric neurons [18,21,26,27]; 2—regulation of synaptic transmission and intestinal motility [18,21,26]; 3—maintenance of the gut and body’s homeostasis [18,21,26,27]; 4—ability to detect signals from immune cells to respond to inflammatory stimuli [16]. 3 5—regulation of oxidative stress to contribute to glutathione synthesis [18]. Additionally, functions currently under study are indicated by dashed arrows: 1—modulation of T and B-lymphocyte activation [16]; 2—ability to suppress T-lymphocyte proliferation [27]. All these functions may interfere with neurodegeneration in general, and the development of Parkinson’s disease in particular. ✓: Confirmed function of EGCs; ¿? need for confirmation of EGC function.

**Table 1 ijms-25-01294-t001:** Types of enteric glia based on their location and functions [18,21,27,28].

Type	Subtype	Location	Functions
Intraganglionar	Myenteric type I	Small and extended star-shaped cells that surround the neurons in the myenteric ganglia	Modulation of enteric neuron activity
			Oxidative stress regulation
			Trophic support
			Neuroinflammation regulation
			Gliogenesis
			Neurogenesis
			Mucosal glia replenishment
	Submucosal type I	Associated with neurons within submucosal ganglia	Modulation of secretory neuron activity
Extraganglionar	Interganglionar type II	Located in the interganglionic fiber tracts	They propagate the signal in the glial network
	Mucosa type III	Some follow nerve fibers, while others terminate in the mucosal epithelium	Influence the maturation of epithelial cells
			Potentially modulate immune responses Identification from postnatal development
	Myenteric plexus/submucosal plexus type III	Located in the extraganglionic regions at the level of the myenteric and submucosal plexuses	Unknown
	Intramuscular type IV	Associated with nerve fibers in the circular and longitudinal muscle layers of smooth muscles	Unknown

**Table 2 ijms-25-01294-t002:** Types of enteric glia based on their reactivity state: activated, reactive, and dysfunctional [18,27,29,30].

Type	Characteristics	Mechanisms of Action
Activated	Controls the activity of surrounding cells	The enteric glial activation encoded by intracellular Ca^2+^ responses modulate enteric excitatory motor and secretomotor neurocircuits
	Exerts beneficial homeostatic effects	
	Responds to physiological stimuli	
Reactive	Responds to physiopathological disturbances of any severity	Responds to intestinal inflammation.
		Contributes to neuronal death during acute intestinal inflammation
	Changes can alter glial activities through gain or loss of functions, which can be beneficial or detrimental	Contributes to vagal anti-inflammatory effects on resident intestinal immune cells after intestinal injury
Dysfunctional	Dysfunctional or maladaptive response of glial cells	Altered enteric glial networks, displaying dysfunctional responses in patients with different GI disorders, including IBD, immunological disorders of the gut or PD
	Exerts harmful effects contributing to a disease, in addition to being permanent	

Abbreviations: GI, gastrointestinal; IBD, inflammatory bowel disease; PD, Parkinson’s disease.

**Table 3 ijms-25-01294-t003:** Enteroglial markers [18,21,26,27,31,32,33,34,35].

Marker	Characteristics	Functions
Nuclear transcription factor (Sox-10)	Key to the development of the neuronal crest cells and the enteric glia	Crucial role in neuronal crest cells and peripheral glia differentiation and maintenance
	Specific marker for EGCs progenitors	Controls and modulates the expression of several key genes for early ENS development
	Found in glial precursors and in most of the mature and immature EGCs	Promotes the expression of various transcription factors crucial for neuronal differentiation, such as *Phox2b* and *Ascl1*
Glial fibrillary acidic protein (GFAP)	It is found along neuronal plexuses.	There is an increased GFAP intensity when the tissue is inflamed or next to colonic cancer
	Does not occur prenatally	
	All subtypes of enteric glia within the mouse ileum express GFAP, but at different levels.	
	Dynamic expression that varies depending on the glial state	
	GFAPκ is the main isoform in colonic EGCs relative to GFAPα and GFAPδ	
Calcium-binding protein (S100β)	Expressed by both progenitors and differentiated enteric glial cells	Among other functions, this protein contributes to structural support and regulation of the immune response
Proteolipid protein 1 (PLP1)	In adult mice, it is expressed in both ENS plexuses, in both, the small and large intestine	Unknown

Abbreviations: EGCs, enteric glial cells; ENS, enteric nervous system; GFAP, glial fibrillary acidic protein; PLP1, proteolipid protein 1.

**Table 4 ijms-25-01294-t004:** Studies in cell cultures and animal models of Parkinson’s disease related to EGCs.

PD Induction	Animal	Findings	Ref
Rotenone	C57BL mice	Increased MT levels.	[65]
		EGCs activation (GFAP-positive)	
	MT KO mice	Severe myenteric neuronal damage	[65]
		Reduced EGCs activation	
		Aggravation of lipid peroxidation	
	C57BL mice	Increased GFAP-IR in the myenteric plexus	[66]
		Activated EGCs before neurodegeneration in the CNS	
		Lack of activation of EGCs at early stages	
	C57BL mice	No MT presence in the intestine of mice	[67]
	Cell cultures	CA or CGA prevented rotenone-induced downregulation of MT in cultured cells	
	C57BL mice Cell cultures	Increased GFAP staining in the myenteric plexus	[68]
		Impaired mitochondrial bioenergetics	
		Activation of inflammatory pathways	
	C57BL/6J mice	TLR4-mediated gut inflammation	[55]
	TLR4-knockout mice	GFAP staining was unaffected by rotenone administration	
	C57BL Mice	Restraint stress exacerbated rotenone-induced activation of EGCs	[69]
	C57BL/6NCrl mice	A TDO inhibitor decreased rotenone-induced labelling of GFAP	[70]
MPTP and MPTP/p	*Macaca mulatta*	No differences in the number/phenotype of Sox-10 IR EGCs	[71]
		EGCs are not a primary target of MPTP in the ENS	
		The ratio of EGC to neurons was decreased by MPTP	
		Less protection of myenteric neurons	
	Marmoset	EGCs IR to Sox-10 were increased	[72]
		MPTP treatment led to inflammation of the ileum	
	C5BL/6 mice	Chronic MPTP/p increased aggregated and nitrated α-syn in GFAP-IR EGCs	[73]
		Acute, elevated 4-HNE in the EGCs after 3 h	
		EGCs could be initial contributors to synucleinopathies in the stomach	
6-OHDA	Sprague-Dawley rats	GFAP IR of EGCs was increased	[74]
	Sprague-Dawley rats	Density of S100β IR EGCs was increased	[75]
		Nigrostriatal neurodegeneration leads to an increased presence of EGCs in the mucosa	
	C57BL/6 male mice	Dual response of EGCs: can promote inflammation or intestinal tissue protection	[76]
Virus AVV	Sprague-Dawley rats	No changes in EGCs in the ileal submucosal plexus	[77]
		Increase in glial number in the myenteric plexus	
		Voluntary running protected from increased EGCs	
A53 α-syn mouse model	Mutant mice expressing human A53T	Increased GFAP IR EGCs in the mucosa, submucosa and myenteric plexus at 3 months of age	[78]
		Co-localization of GFAP-IR EGCs and TLR2 IR	

Abbreviations: 4-HNE, 4-hydroxynonenal; 6-OHDA, 6-hydroxydopamine; α-syn, α-synuclein; AVV, adeno-associated virus; CA, caffeic acid; CGA, chlorogenic acid; CNS, central nervous system; EGCs, enteric glial cells; GFAP, glial fibrillary acidic protein; IR, immunoreactivity, immunoreactive; KO, knockout; MT, metallothioneins; MPTP, 1-methyl-4-phenyl-1,2,3,6-tetrahydropyridine; MPTP/p, MPTP/probenecid; PD, Parkinson’s disease; TDO, tryptophan 2,3-dioxygenase; TLR, Toll-like receptor.

**Table 5 ijms-25-01294-t005:** Studies related to EGCs performed in samples of Parkinson’s patients.

Number of Samples	Samples	Findings	Refs
Control and PD patients (2 samples each)	Ascending Colon	GFAP and Sox-10 were significantly elevated	[63]
		No significant changes in the S100β marker	
		Levels of glial markers are negatively correlated with disease duration	
24 PD, 6 progressive supranuclear palsy and 6 multiple system atrophy patients	Colonic biopsies	Hypophosphorylation of GFAP in EGCs during PD	[86]
		GFAPκ was the major isoform in colonic EGCs	
19 asymptomatic PD patients	Colonic biopsies	Increase in S100β-positive glial cells	[87]
		Activation of enteric glial cells	
		Abnormal tissue repair with development of fibrosis in the mucosa of PD patients	
128 patients with PD	Serum	GDNF may act as a protective factor in the prevention of constipation	[88]
		Reduction in GDNF affects the integrity of intestinal mucosal barrier	
16 Lewy’s body, 12 non-Lewy’s body disorders cases	Colonic biopsies	Enteric glial cells express LRRK2	[89]
		EGC-expressed LRRK2 could participate in the modulation of intestinal α-syn aggregation and inflammation in the gut	
18 patients with advanced PD, 4 untreated patients with early PD	Duodenal biopsies	Increased size and density of GFAP-positive EGCs suggesting reactive gliosis	[90]
		No colocalization between markers α-syn-5G4 and GFAP antibodies	

Abbreviations: α-syn, α-synuclein; EGCs, enteric glial cells; GDNF, glial-derived neurotrophic factor; GFAP, glial fibrillary acidic protein; LRRK2, leucine-rich repeat kinase 2; PD, Parkinson’s disease.

## Data Availability

Not applicable.

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
