# Peer review of "From the Gut to the Brain: The Role of Enteric Glial Cells and Their Involvement in the Pathogenesis of Parkinson’s Disease"

_ijms, 2024, doi:10.3390/ijms25021294_

Round 1

Reviewer 1 Report

Comments and Suggestions for Authors

Overall, this is a well written review publication and offers a good overview of the role of glial cells in pathogenesis of Parkinson’s disease. This manuscript shows rich content, providing a deep insight for some works. The review is within the journal’s scope, and I found it to be well-written, providing sufficient information. However, before publication some points need to be clarified.

My comments:

Line 81 and the rest of the text – I have the impression that the authors consider gastrointestinal system and gastrointestinal tract to be synonymous. However, they are in fact two different anatomical structures.

Line 219 and the rest of the text – The authors should ensure that they use term “expression” in relation to gens only.

Line 100 – the authors should follow journal references style guidelines. In the whole manuscript, please avoid sentence construction like “the studies by Li et al. [4] and Kim et al. [11]”. Authors names should be replaced with appropriate reference numbers.

Line 122 - in the main text SIBO acronym is used only one time. I see no sense to abbreviate it. 

Line 138 – the authors should emphasize that the submucosal plexus is more prominent in the small and large intestine, than in the stomach.

Line 155 – please provide short methodology of this review.

Line 230 - in the main text LPS acronym is used only one time. I see no sense to abbreviate it. 

Reviewer 2 Report

Comments and Suggestions for Authors

This manuscript reviews the topic of how enteric nervous system (ENS)  influences neurodegenerative disease processes in the brain. Specifically, the role of enteric glial cells (EGCs) in neurons-immune interactions is emphasized. While focus has been primarily on enteric neurons for neurodegenerative processes, evidence from clinical samples, in vitro and animal models are consistent with the activation of these abundant cells correlate with brain neurodegeneration associated with Parkinson's disease (PD), including potentially triggering synucleinopathies.

The review is well-written (but “running” is misspelled on line 820). To improve completeness, however, the following modifications are recommended.

1.    While the reviewers point out that EGCs express Toll Like receptor 4, the implication of LPS in the gut in activation of the cells is not covered. This would be a good opportunity to include consideration of how bacterial infection or microbiota composition may influence PD initiation. 

2.    The reviewers also point out potential interventions, such as enzyme inhibitors, nutraceuticals and physical exercise, are there other, more speculative targets that could be described?

3.    The review would benefit from inclusion of a figure to summarize the known and speculative functions of EGCs in neurodegeneration. 

4.    While associations are compelling between the ENS and neurodegeneration, the authors should consider evidence that fail to support a direct causal link between EGCs function and disease. 
